# Evaluation of the Analgesic Efficacy of Undiluted Intraperitoneal and Incisional Ropivacaine for Postoperative Analgesia in Dogs after Major Abdominal Surgery

**DOI:** 10.3390/ani13091489

**Published:** 2023-04-27

**Authors:** Inken S. Henze, Victoria Navarro Altuna, Joëlle I. Steiger, Paul R. Torgerson, Annette P. N. Kutter

**Affiliations:** 1Section of Anaesthesiology, Vetsuisse Faculty, University of Zurich, 8057 Zurich, Switzerland; victoria.navarroaltuna@uzh.ch (V.N.A.); akutter@vetclinics.uzh.ch (A.P.N.K.); 2Section of Epidemiology, Vetsuisse Faculty, University of Zurich, 8057 Zurich, Switzerland; paul.torgerson@uzh.ch

**Keywords:** anaesthesia, analgesia, canine, companion animals, laparotomy, locoregional, pain, visceral surgery

## Abstract

**Simple Summary:**

Pain in animals is an important problem as a lack of verbal communication might underestimate their perception and thus lead to undertreatment. Abdominal surgery is frequently performed in dogs, be it elective or for curative reasons. Recommendations to administer a local anaesthetic agent (i.e., a drug that inhibits pain at the site where it is administered) into the abdomen are existing without knowing how good they really work to prevent pain after surgery. In this study, pain, sedation level, the heart rate and opioid requirements were assessed in dogs that underwent abdominal surgery for an underlying disease. Half of the dogs received the investigated drug ropivacaine, while the other half received saline. Pain and sedation level were assessed using scientifically established scores. Rescue analgesia was provided if necessary. Sedation, pain on one score, and sensitivity to pressure next to the surgical wound were not different between both groups. Another pain score achieved slightly higher scores in dogs treated with ropivacaine, and the heart rate was lower in this group. These findings lead to the conclusion that the dosage and concentration used in this study should not be investigated further and cannot be promoted for clinical use.

**Abstract:**

Recommendations for intraperitoneal (IP) and incisional (INC) administration of local anaesthetics after visceral surgery exist, but evidence is scarce. This prospective, randomized, blinded, controlled, clinical trial compared postoperative pain in dogs undergoing major abdominal surgery. Sixteen client-owned dogs were anaesthetized with a standardized balanced protocol including opioids and received either 2 mg/kg ropivacaine IP (0.27 mL/kg) and a 1 mg/kg INC splash (0.13 mL/kg) or equal volumes of saline. Influence of the treatment on heart rate (HR) and postoperative pain was assessed using the Short Form of the Glasgow Composite Pain Scale (GCPS-SF), a dynamic interactive visual analogue scale (DIVAS) and mechanical nociceptive threshold testing (MNT). Data was tested with mixed ordinal regression and log linear mixed models for 0.5, 1, 2, 3, 4, 6, 8, 10 and 12 h after extubation. Rescue analgesia was given to 3/8 dogs after ropivacaine and 0/8 dogs after saline. GCPS-SF and MNT were not different between groups. DIVAS was slightly higher after ropivacaine (odds increased by 5.44 (confidence interval (CI) 1.17–9.96, *p* = 0.012)), and HR after ropivacaine was 0.76 * that after saline (CI 0.61–0.96, *p* = 0.02) with no effect of time (*p* = 0.1). Undiluted ropivacaine IP and INC was not beneficial for postoperative analgesia.

## 1. Introduction

Abdominal surgery is commonly performed in dogs. Although it is widely accepted that it results in moderate to severe postoperative pain, the analgesic regimen varies greatly, even within a small country like Switzerland [1]. As animals cannot verbally communicate any perception of pain, and are often discharged early in the postoperative period, the risk of insufficient analgesia is high. To support a quick and smooth postoperative recovery, perioperative analgesia plays an important role [2]. It should be multimodal and impede nociception and pain as much as possible.

Local anaesthetic agents are the only class of drugs that can prevent pain. In contrast, systemically administered analgesics only attenuate pain to different levels, depending on their analgesic properties. Despite this fact, a query among Swiss veterinarians demonstrated that locoregional anaesthesia was employed by 43.8% of the participating veterinarians only, with most of them limiting their application to the subcutaneous administration of the short-acting local anaesthetic lidocaine. Long-acting local anaesthetics like ropivacaine or bupivacaine were used by only a small minority of veterinarians (7.7%) [1].

Studies evaluated the intraperitoneal (IP) administration of bupivacaine and levobupivacaine and some demonstrated reduced postoperative pain in dogs [3,4]. When compared with bupivacaine, ropivacaine led to shorter postoperative analgesia [5]. In healthy dogs undergoing ovariohysterectomy, 1 mg/kg ropivacaine IP failed to demonstrate a beneficial effect on postoperative pain, compared to a control group [6]. Compared with bupivacaine, ropivacaine possesses less cardiotoxic potential [7,8], and ropivacaine is lower-priced compared with levobupivacaine, both factors which could make ropivacaine a favourable choice for clinical implementation. As newer, more sophisticated techniques like the transversus abdominis plane block [9] or the quadratus lumborum block [10] require advanced knowledge, skills and equipment, they are currently not feasible in many places like small veterinary practices, shelters, or trap-neuter-release programs. The Global Pain Council of the World Small Animal Veterinary Association (WSAVA-GPC) recommends IP and INC administered local anaesthetic agents for visceral surgery in dogs and cats [11,12]. As to date evidence is little, those recommendations are mainly consensus-based, and the WSAVA-GPC recommends more clinical studies evaluating their effect. Especially easy-to-use techniques that do not need any advanced training or equipment might present a sensible additional component to any analgesic strategy for veterinary patients undergoing abdominal surgery. In places with limited access to drugs or financial restrictions, patient well-being could be significantly enhanced as local anaesthetics are comparably low-priced and do not underly any advanced legal regulations. Furthermore, veterinarians have been challenged by the current opioid crisis and the COVID-19 pandemic, facing shortages especially in the availability of full µ-receptor agonists [13].

Approaches to measuring pain in animals are ideally objective and validated. While scales for pain assessment like the Short Form of the Glasgow Composite Pain Scale (GCPS-SF) have been validated for assessment of acute pain in dogs [14], pain scoring generally remains observer-dependent and, to a certain degree, subjective [15]. Mechanical nociceptive threshold testing (MNT) has been used as a more objective approach to assessing postoperative pain, yet dogs developed a learning behaviour in response to the anticipated stimulus [16]. To the authors’ best knowledge, no study has been published in clinical canine patients undergoing major abdominal surgery, in which IP and INC administered ropivacaine was evaluated for its postoperative analgesic effect. Therefore, this study aimed to show that dogs treated with ropivacaine IP and INC achieve lower pain scores in the first 12 postoperative hours than dogs in a control group. The study’s hypothesis was that especially in the first six postoperative hours, dogs treated with ropivacaine would be less painful than dogs treated with saline.

## 2. Materials and Methods

Ethical approval for this blinded, randomized, prospective, controlled clinical study was granted by the ethical committee of the canton Zurich (ZH104/20). For each patient, written informed owner consent was obtained before inclusion in the study.

### 2.1. Animals

Client-owned dogs of either sex undergoing anaesthesia for abdominal surgery were included in the study. Fasting time was at least six hours or less in emergency procedures. Pre-established exclusion criteria included aggression and increased stress levels on handling or dogs under long-term analgesic treatment for any condition. Healthy bitches undergoing elective ovariectomy or ovariohysterectomy, and pregnant ones presenting for caesarean section were also not considered for participation. Breed, age, sex, and bodyweight were recorded. The dogs were randomly assigned to either the ropivacaine (R) or the control (C) group using a computer programme “www.randomization.com (accessed on 10 May 2021)” before the start of the study.

### 2.2. Anaesthesia

Based on the medical history and preanaesthetic clinical examination, an American Society of Anesthesiologists (ASA) status was assigned to each dog. All cases were managed by the same anaesthetist. If intramuscular (IM) sedation was required for aseptic placement of an intravenous (IV) catheter of appropriate size (VasoVet, B. Braun Medical AG, Seesatz 17, 6204 Sempach, Switzerland), 4–5 µg/kg dexmedetomidine (Dexdomitor, 0.5 mg/mL, Provet AG, Gewerbestrasse 1, 3421 Lyssach, Switzerland) was injected. The anaesthetic regimen consisted of 2–5 µg/kg fentanyl (Fentanyl Sintetica, 50 µg/mL, Sintetica SA, Via Penate 5, 6850 Mendrisio, Switzerland) IV, titrated to effect, combined with 0–5 µg/kg dexmedetomidine IV for premedication. Anaesthesia was induced with 0.5 mg/kg propofol (Propofol 1% MCT Fresenius, Fresenius Kabi, Am Mattenhof 4, 6010 Kriens, Switzerland) IV titrated to effect with additional boluses of 0.25 mg/kg, or 0.25 mg/kg alfaxalone (Alfaxan Multidose, 10 mg/mL, Dr. E. Graeub AG, Rehhagstrasse 83, 3018 Bern, Switzerland) IV titrated to effect with additional boluses of 0.25 mg/kg. A co-induction was performed with 0.1–0.2 mg/kg midazolam (Dormicum, 5 mg/mL, Roche Pharma, Gartenstrasse 9, 4052 Basel, Switzerland) if no portosystemic shunt was present. The airway was secured with a cuffed endotracheal tube of appropriate size and general anaesthesia was maintained with sevoflurane in oxygen and air (initial FIO_2_ 50%, concentration adjusted as necessary) via a circle system (Aespire View, Anandic, Stadtweg 24, 8245 Feuerthalen, Switzerland). A fentanyl infusion (1–10 µg/kg/h) was given for intraoperative analgesia. Additionally, one dog (N° 5) received 4 mg/kg carprofen (Rimadyl, 50 mg/mL, Zoetis Schweiz GmbH, Rue de la Jeunesse 2, 2800 Delémont, Switzerland) IV every 24 h, and the other dogs received 30 mg/kg metamizole (Minalgin, 500 mg/mL, Streuli Tiergesundheit AG, Bahnhofstrasse 7, 8730 Uznach, Switzerland) IV, repeated after eight hours. All dogs were mechanically ventilated with a pressure-controlled synchronized intermittent mandatory ventilation mode set to obtain a tidal volume of 10 mL/kg and a respiratory rate set to obtain end-tidal CO_2_ (EtCO_2_) between 35 and 45 mmHg. Immediately following induction of general anaesthesia, the cardiorespiratory monitoring was installed in each patient comprising multi-gas analysis, capnography, ECG, pulse oximetry, non-invasive blood pressure and core body temperature measurement with a multi-parameter monitor (Cardiocap/5, GE Datex-Ohmeda, Anandic, Stadtweg 24, 8245 Feuerthalen, Switzerland). Acetated Ringer’s solution (Ringer Acetat “Bichsel”, Grosse Apotheke Dr. G. Bichsel AG, Bahnhofstrasse 5a, 3800 Interlaken, Switzerland) was started at an infusion rate of 5 mL/kg/h. The anaesthetic inhalant concentration was titrated to the appropriate depth based on clinical signs. Body temperature was maintained using a circulating warm water mattress (HICO-AQUATHERM 660, Nufer Medical AG, Morgenstrasse 148, 3018 Bern, Switzerland) and a forced air warming system (Mistral-Air, Anandic, Stadtweg 24, 8245 Feuerthalen, Switzerland).

Surgeries were performed by an experienced surgeon. They all included coeliotomy via midline incision. Before complete closure of the linea alba, the dogs in group R received 2 mg/kg undiluted ropivacaine (Ropivacain Sintetica 0.75%, Sintetica SA, Via Penate 5, 6850 Mendrisio, Switzerland) IP, splashed into the abdominal cavity at the cranial end of the incision by the surgeon. The tip of the syringe containing the drug was inserted before the last sutures were performed. To ensure a distribution inside the abdomen, it was directed cranially, caudally, and laterally to both sides while the drug was slowly injected. The dogs in group C received an equal volume (0.27 mL/kg) of isotonic saline solution (NaCl 0.9%, B. Braun Medical AG, Seesatz 17, 6204 Sempach, Switzerland). Once the linea alba was closed, the dogs in group R received an additional splash of 1 mg/kg undiluted ropivacaine onto the muscular and subcutaneous layers of the incision line (INC) while the dogs in group C received an equal volume (0.13 mL/kg) of an isotonic saline splash. The anaesthetist, the surgeon and the person performing the postoperative monitoring were all blinded to any group allocation and to the drug used. General anaesthesia was discontinued after the skin was completely sutured. The length of the incision line and the distance between the xiphoid and the os pubis were measured and recorded, and their ratio calculated. The responsible surgeon was asked for an estimation of his or her anticipated pain intensity for the respective surgical intervention on a numerical rating scale (NRS) ranging from zero to ten, with zero reflecting the absence of any pain, and ten reflecting the worst pain imaginable. Every dog received 4 mg/kg pethidine (Pethidin Streuli, 50 mg/mL, Streuli Tiergesundheit AG, Bahnhofstrasse 7, 8730 Uznach, Switzerland) IM when the fentanyl infusion was stopped. The dogs were transported to the recovery ward and extubated once the swallowing reflex had recurred. One dog in group R received 0.5 µg/kg dexmedetomidine IV because of a very rough recovery (N° 15). The anaesthetic and analgesic protocol is displayed in Figure 1.

### 2.3. Postoperative Measurements

Postoperative measurements and assessments were performed by one single observer (JS) 30 min (T0.5), and one (T1), two (T2), three (T3), four (T4), five (T5), six (T6), eight (T8), ten (T10) and 12 (T12) hours after extubation. Heart rate (HR) was counted by auscultation at the same time points. Sedation was assessed using a 7-item scale published by Grint et al. [17] and validated by Wagner et al. [18]. Pain scoring was performed using GCPS-SF, a dynamic interactive visual analogue scale (DIVAS), and MNT. On GCPS-SF, the achieved score out of 24 (or 20 if mobility could not be assessed) was noted. For DIVAS, a 100 mm line was used with its left end referring to the absence of any pain and its right end referring to the worst possible pain. To measure MNT, pressure with the force algometer was steadily increased in three locations at ten millimetres around the incision line until the dog showed any kind of response. Possible responses included a sudden movement, turning the head towards the device, a sudden tense abdomen, growling, flinching, crying, snapping, or attempts to bite. At any response, pressure was immediately released, and the applied force (in N) was noted (force range 0.5–25 N, accuracy ± 0.5 N according to the manufacturer). The mean of the three measurements was calculated and used for statistical analysis. In all dogs, assessments and measurements at every time point were performed in the same order. First, assessments performed from outside the kennel were done, followed by items from the sedation scale and GCPS-SF involving interaction with the dog. MNT measurements completed the assessment rounds. All scorings were video recorded and assessed by a second, independent investigator (VNA), who was also blinded to the treatment. Rescue analgesia (0.2 mg/kg methadone IV; Methadon Streuli, 10 mg/mL, Streuli Tiergesundheit AG, Bahnhofstrasse 7, 8730 Uznach, Switzerland) was administered if >5/24 or >4/20 were reached on the GCPS-SF. The dog was re-assessed 20 min later and received another 0.2 mg/kg methadone if the score was still >5/24 or >4/20. The time to first administration of methadone and the total amount of methadone per patient were recorded within the first twelve postoperative hours.

### 2.4. Statistical Methods

Randomization was done for 16 dogs with an online tool. After the measurements, data was entered in an Excel sheet (Microsoft Office 2019, Microsoft Corporation, Redmond, WA, USA) and statistical analysis was performed. To prevent an influence of the administered methadone, data after administration of rescue analgesia was removed from statistical analysis. Data recorded only once was tested for normal distribution using a Kolmogorov-Smirnov test, and compared with Student’s *t*-test or Mann-Whitney test using GraphPad Prism 9.2.0 (GraphPad Software Inc., San Diego, CA, USA). Data of both groups measured at repeated time points were compared using mixed ordinal regression for analysis of the effects of the local anaesthetic agent on the pain score, and a log linear mixed models for the effect on HR. A *p*-value < 0.05 was considered significant. These analyses were undertaken in R (R Core Team (2022). R: A language and environment for statistical computing. R Foundation for Statistical Computing, Vienna, Austria; URL https://www.R-project.org/ accessed on 10 May 2022).

## 3. Results

A total of 16 dogs were recruited for the pilot study. No dog needed to be excluded. Each group consisted of eight dogs. No difference was detected between groups for age, bodyweight, sex, ASA status, duration of anaesthesia, duration of surgery, the NRS score reflecting the surgeon’s anticipated pain intensity of the respective surgical intervention, and ratio of length of incision to xiphoid-to-pubis distance (Table 1).

The premedication of the dogs, the induction agents and the procedures are summarized in Table 2. One dose of rescue methadone (=0.2 mg/kg total dose each) was administered to 3/8 dogs in group R once at T2, T3 or T10, respectively. No dog in group C received rescue methadone.

The GCPS-SF was not affected by treatment (*p* = 0.46) or time (*p* = 0.2, Figure 2a). The DIVAS scores were higher in group R with the odds increased by 5.44 (confidence interval (CI) 1.17–9.96, *p* = 0.012), with no effect of time (*p* = 0.24, Figure 2b). For the MNT scores no effect of treatment (*p* = 0.27), but a decrease with time was observed. The odds of the MNT scores decreased by a factor of 0.74 (CI 0.64–0.86) for each hour (*p* < 0.01, Figure 2c).

Due to insufficient video quality, the recordings could not be completely evaluated for each dog and each timepoint, and thus could not be compared to the live scorings. It was not possible for the independent assessor to confidently assign GCPS-SF scores to the dogs based on the video recordings only.

The sedation score was unaffected by treatment (*p* = 0.31) but increased with time (odds ratio for each unit increase in score was 1.15 per hour (CI of odds ratio 1.096–1.215), *p* < 0.01).

Group R had a HR of 0.76 * that of group C ((CI 0.61–0.96), *p* = 0.02) with no effect of time (*p* = 0.1).

## 4. Discussion

To the authors’ knowledge, this study is the first to evaluate postoperative analgesia after an IP and INC splash with undiluted ropivacaine 0.75%, compared to a control group, in dogs undergoing major abdominal surgery. In contrast to our hypothesis, one of the assessed pain scores (DIVAS) was higher in group R, and more dogs in group R received rescue analgesia than in group C. Similar to the results of a former study comparing the addition of INC bupivacaine to IP administration alone [19], the combination of ropivacaine INC and IP in the current study did not show to have a positive effect on postoperative pain scores. In former studies in dogs, IP ropivacaine without additional INC administration was evaluated at doses of 1 mg/kg [6] and 3 mg/kg [5,20] against a control group regarding pain scores and required rescue analgesia. In agreement with the current study, no differences in pain scores between treatment and a control group had been found.

### 4.1. Study Design

Possible reasons for a lack of differences in GCPS-SF scores between treated and untreated animals are numerous. The first and often ignored reason could be an underpowered study. Originally, the current pilot study was planned to collect enough data in these first 16 animals to define the sample size needed to show a benefit in group R. However, the fact that GCPS-SF scores exceeded 5/24 (4/20) in dogs of group R only made the current data unsuitable for sample size calculation, and further examination of the present dose, volume, and concentration should be abolished. Another reason could be the different types of surgery included in this study. While all surgeries included coeliotomy via midline incision which leads to somatic pain, the visceral component should not be neglected. The absence of a difference in the ratio of the length of incision to the xiphoid-to-pubis distance between the groups only refers to the somatic component of pain. In our small sample size, we did not see any of the repeated procedures (as listed in Table 2) to be especially painful. A large prospective study in humans evaluated postoperative pain intensity (NRS scores) between different surgeries [21]. Interestingly, when compared to the cohort in our study, those dogs that received rescue analgesia were not those undergoing interventions that were rated as especially painful by human patients. Of the surgeries performed in our study, cholecystectomy was rated highest with a mean NRS score of 5.83, closely followed by splenectomy (5.56) and small bowel resection (5.45). The dogs that received rescue analgesia underwent closure of portosystemic shunt, adrenalectomy, and splenectomy. Adrenalectomy was rated quite low in human patients (mean NRS score 3.86). Regarding splenectomy, one of three dogs undergoing this procedure required rescue analgesia while the other two did not. In a follow-up study with an amended study design, a larger cohort will be necessary to compensate for the difference in surgical procedures.

### 4.2. Local Anaesthetic Protocol

A lack of effect could also be explained by an insufficient distribution to the site of effect due to a too low dose, volume, or concentration. The volume administered in the current study by using undiluted ropivacaine 0.75% at a dose of 2 mg/kg IP was only 0.27 mL/kg. In other studies, total volumes of ropivacaine from 0.6 mL/kg [5] to 1.2 mL/kg [20] were administered. Screening the available literature, several dilution regimens have been used for different local anaesthetic agents used for IP administration in dogs [3,4,5,6,19,20,22,23]. The reported volume instilled into the peritoneal cavity in the different investigated local anaesthetics ranges from 0.5 to 1.76 mL/kg in dogs, yet it remains unclear which volume is ideal for an adequate peritoneal surface anaesthesia. One might argue that a larger volume will lead to a larger peritoneal surface covered by the local anaesthetic solution, which may then target a higher number of free nerve endings. On the other hand, a lower concentration because of a higher dilution might have a smaller effect on the peritoneal neurons. No study has been conducted on pharmacokinetics of ropivacaine after IP administration in dogs, which makes a definition of the maximum dose that can be safely administered IP difficult. As in the present study investigations were conducted on clinical patients, and to limit the risk of overdose, the authors did not intend to exceed a maximum dose of 3 mg/kg, leading to an IP dose of 2 mg/kg combined with an INC dose of 1 mg/kg. In adult humans, IP instillation of 20 mL ropivacaine 0.25% or 0.75% was shown to have a similar pharmacokinetic profile to extravascular administration, and no signs of clinical toxicity were noticed [24]. Signs of toxicity in the current study could have been recognized by abnormal behaviour (e.g., increased sedation scores) or negative effects on the cardiovascular system, such as tachycardia, bradycardia, or arrhythmias.

### 4.3. Pain Scores: GCPS-SF and DIVAS

In the present study, two pain scoring systems were used. The GCPS-SF is believed to be a more objective method as the observer needs to select exactly from predetermined criteria and has therefore been used as the threshold to administer rescue analgesia. DIVAS is a subjective scale also involving interaction with the animal [25]. In DIVAS, the need to choose from set criteria at every individual timepoint is absent, and the observer is allowed to involve a subjective assessment of pain development over time. In this study, DIVAS scores were slightly higher in group R. One explanation for this difference to GCPS-SF might be the more subjective character of DIVAS, detecting slight hints for pain that might remain undetected by the categories on GCPS-SF.

Nevertheless, the higher DIVAS scores in group R together with the fact that only dogs of group R received rescue methadone contradicts the hypothesis that dogs in group R would be less painful than dogs in group C. The small but significant difference between DIVAS scores in favour of group C possibly shows that eight dogs per group were sufficient to show that the examined protocol did possibly cause more harm than benefit. If the administration of an acid drug might cause abdominal discomfort by irritating the peritoneum without inducing sufficient desensitization cannot be answered with the current study but must be taken into consideration. Intraperitoneal pH in dogs has been shown to be around 8.03 ± 0.13 (mean ± SD) immediately after the abdominal cavity was incised [26]. Although acid injected into the peritoneal cavity has been rapidly neutralized in dogs [27], an initial irritant or nociceptive effect on the peritoneum through the instillation of the acidic ropivacaine solution cannot be ruled out.

### 4.4. MNT in Dogs

To objectify the pain scoring further, the use of MNT has been suggested and proved feasible [28]. No difference could be found in MNT values between the groups, but a decrease with time was observed. Lower MNT values after surgery compared to baseline have been reported before [29]. The fact that over time, lower MNT values were recorded could either be explained by learning and anticipation by the dogs through the repeated measurements [16], or by an actual increase in pain perception that could not be shown with GCPS-SF and DIVAS. The decrease in MNT values over time could also be influenced by termination of action of ropivacaine in one group before MNT measurements were stopped. Although no direct effect of treatment could be shown on MNT (*p* = 0.27), a highly significant interaction (*p* < 0.01) supporting an association between time and group was demonstrated. Another possible influencing factor could be the fact that sedation scores and GCPS-SF assessments including palpation of the region around the surgical wound preceded MNT testing, which was regarded as the most invasive test. Until MNT was performed at the end of any assessment round, the dogs might have been sensitized through the other tests. To minimize any unforeseeable influence, the same order of all tests and assessments was kept in all dogs at all timepoints.

The algometer used in the current study for measuring MNT had a blunt tip with a diameter of 2 mm, which is in accordance with the manufacturer’s recommendation for use in canines. Nevertheless, in the two other studies using a hand-held algometer from the same manufacturer for assessment of postoperative pain, a 1 cm rounded tip [6] or a 4 mm probe tip [5] were used. Gomes et al. [20] used an electronic device with unknown tip size. It has been demonstrated that a smaller diameter probe tip required lower forces to generate a response but generated higher pressures [30,31,32]. This difference in methodology makes a comparison difficult, but also in the mentioned studies in dogs, no differences between groups could be shown.

### 4.5. Sedation Scores

Sedation scores did not differ between groups, so a neurological side effect of ropivacaine seems unlikely. On the other hand, HR was around 25% lower in group R. Increased sympathetic stimulation in the control group as an explanation for higher HR is not supported by the pain score findings. Systemic absorption of ropivacaine does occur via the peritoneum, therefore a lower HR can be explained by the ability of local anaesthetics to slow conduction velocities in the heart [8]. This finding must be considered in future dose researching studies as plasma levels are not easily measurable in clinical studies. Possible neurological toxicity signs due to systemic absorption could be masked by sustained sedation after anaesthesia and might only become obvious well after recovery.

Interestingly, sedation scores increased over time. One explanation in this study could be the advanced time of the day. Approaching the evening, our hospital becomes quieter, fewer people are around and the animals tend to fall asleep more easily than during the day. Another explanation might be the familiarisation with both the staff and the environment with time when the animals get more used to their situation. This contrasts with the increased skin sensitivity over time measured with MNT in the current study. It seems that MNT increases could be measured although sedation scores were increasing.

### 4.6. Limitations

#### 4.6.1. Analgesic Protocol

Several limitations are present in the current study. Although attempts had been made to standardize the anaesthetic protocol for all dogs, this was not possible in a very strict manner due to the clinical nature of the study and the underlying pathologies of the dogs included. The use of dexmedetomidine and fentanyl to effect might have influenced the individual development of nociception before administration of the study drugs. While the dexmedetomidine dose used for IM sedation was estimated based on the clinical experience of the anaesthetist, both dexmedetomidine and fentanyl administered IV were titrated to effect. The goal was to achieve visible sedation before induction of general anaesthesia. While the same dose would be preferable in all dogs, the chosen approach with individualized doses was thought to be more ethical and justified for clinical patients. Dexmedetomidine administered either IM or IV could possibly have an influence on the analgesic protocol. Nevertheless, the dosage used (up to 5 µg/kg) is comparably low. At a dose of 5 µg/kg IV, terminal-phase half-life for dexmedetomidine in dogs was shown to be 36 ± 6 min [33]. After IM administered dexmedetomidine at a dose of 10 µg/kg, mean half-life was 25.5 (11.5 to 41.5) minutes [34]. In the current study, surgery hadn’t started until 36 min after IV, or until 41 min after IM injection of dexmedetomidine in any of the dogs. The latter data is for twice or more than twice the dose used in our study. Still, any alterations on the molecular level provoked by dexmedetomidine and leading to changes in the pain pathway that may be present for a longer time cannot be ruled out.

It is known that pre-emptive administration of local anaesthetics would provide superior analgesia and decrease the influence of other drugs mentioned above. In the current study the local anaesthetic agent was administered after surgical stimulation of peritoneal free nerve endings. Instillation of local anaesthetics both INC and IP right after the opening of the abdomen could lead to better results in future studies, but its effect might be impaired by surgical suctioning or lavage during the procedure.

To provide multimodal analgesia, all dogs received a potent non-opioid analgesic before surgery was started. While only one dog received carprofen (in group R), the other 15 dogs received metamizole. It is possible that this difference might have influenced the outcome further, but that individual dog’s scores were not obviously different to those of the other dogs. The analgesic efficacy of peroral carprofen and metamizole has been compared in dogs after ovariohysterectomy [35]. Both drugs provided comparable postoperative analgesia in that study, and the only statistical significance was demonstrated 30 min after extubation, when GCPS-SF scores were significantly higher in the metamizole group. But since dogs in the carprofen group were profoundly sedated at that time point, the reliability of GCPS-SF in these dogs at that timepoint is questionable. In the present study, all dogs received fentanyl via a continuous rate infusion starting at a rate of 5 µg/kg/h for the duration of surgery. The rates were adapted based on cardiorespiratory variables and ranged between 3 to 10 µg/kg/h in all patients. Since the duration of surgery did not differ between groups, an influence of a prolonged fentanyl infusion on postoperative pain was deemed unlikely.

In the current study, all dogs received pethidine at the end of surgery, when the fentanyl variable rate infusion was stopped. Pethidine, at a dose of 5 mg/kg IM, has been shown to produce reliable analgesia for up to four hours [36], or up to two hours after a dose of 2–3.5 mg/kg IM in dogs [37]. This short-acting µ-opioid-receptor agonist was chosen to provide adequate analgesia in the initial postoperative phase, when remaining sedation or dysphoria could have masked any signs of pain, therefore possibly leading to an undertreatment of present postoperative pain. The fact that two of the three dogs received rescue analgesia at T2 and T3, respectively, could support a relevant analgesic effect of pethidine up to these timepoints. In addition, intraperitoneal and incisional local anaesthetics have been recommended as an adjunct to systemic analgesia only, not as a replacement [12]. Therefore, this study aimed to investigate its analgesic effect when used in a multimodal analgesic protocol and not as the sole analgesic agent.

#### 4.6.2. Factors Influencing Sedation and Pain Scores

Recommendations to use objective scoring systems for assessment of both pain and sedation have been promoted to minimize all potential sources of variation [36]. The 7-item sedation scale used in this study intended to make sedation assessment more objective, and was also used by Gomes et al. [20]. Similar to Lambertini et al. [5] we had the subjective impression that the dogs’ temperament influenced the score achieved on GCPS-SF, in which points need to be given due to anxious or nervous behaviour without considering a dog’s behavioural nature. To date, no evidence could be found for a dog’s temperament to influence its pain behaviour [38]. This is in contrast to findings in humans, in which personality traits like catastrophizing, extraversion and neuroticism have been associated with an increased sensitivity to pain [39]. One explanation might be the different scoring methods between self-reporting in people and observer-reporting in animals. A survey among the general public and veterinarians showed that sensitivity to pain was rated differently based on a dog’s breed [40], a factor that might influence pain scoring in clinical studies involving dogs of a variety of breeds. In addition, the direct assessor of pain was a relatively inexperienced 5th year veterinary student. Postoperative GCPS-SF and DIVAS scores were shown to depend on the observer’s level of experience and were different between veterinarians and veterinary students [15], with the latter assigning higher scores in general. However, in the current study, the quality of the videos was insufficient to compare the video assessments with the direct assessments or to reanalyse the video data for all animals.

### 4.7. Future Perspectives

Local anaesthetic agents administered IP could theoretically be a valuable adjunct for pain therapy after abdominal surgery in dogs [3,5,19,23]. However, evidence of no effect in lower dosages and volumes is growing. To determine a more effective dosage and volume of local anaesthetic drugs for IP/INC administration, more research is needed. In the shadow of the American opioid epidemic, the COVID-19 pandemic, and their subsequent shortages in medical supplies, but also regarding financial restrictions in many places and statuary regulations, local anaesthetic agents could play a vital role in future pain therapy. Easy-to-use techniques that do not require advanced knowledge, skills or equipment could pose a sensible adjunct to systemic analgesia if taught to many veterinarians worldwide. Evidence-based recommendations regarding ideal dosages, concentration and total volumes for IP and INC local anaesthetics should be defined.

## 5. Conclusions

In conclusion, this study did not show that less systemic opioids are needed when undiluted ropivacaine was administered IP in combination with INC. The increased DIVAS and higher incidence of rescue analgesia in group R suggest that the current dosage, concentration, and volume should not be investigated further. As evidence for the optimal dosage and volume of ropivacaine to be instilled into the peritoneal cavity remains unclear, further research is warranted.

## Figures and Tables

**Figure 1 animals-13-01489-f001:**
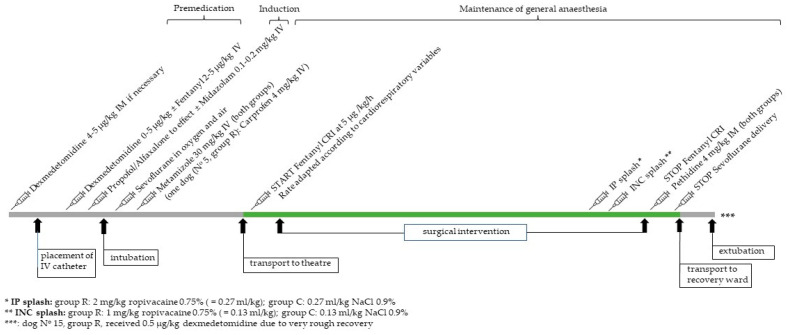
Visualised analgesic protocol and interventions for 16 dogs undergoing major abdominal surgery, and receiving either ropivacaine 2 mg/kg intraperitoneally and 1 mg/kg incisionally (group Ropivacaine), or an equal volume of isotonic saline (group Control), until extubation. Note that the distances on the chart are not true to real time scale. The green bar displays the part of the study the animal spent in the surgical theatre. IM = intramuscular, IV = intravenous, CRI = continuous rate infusion, IP = intraperitoneal, INC = incisional.

**Figure 2 animals-13-01489-f002:**
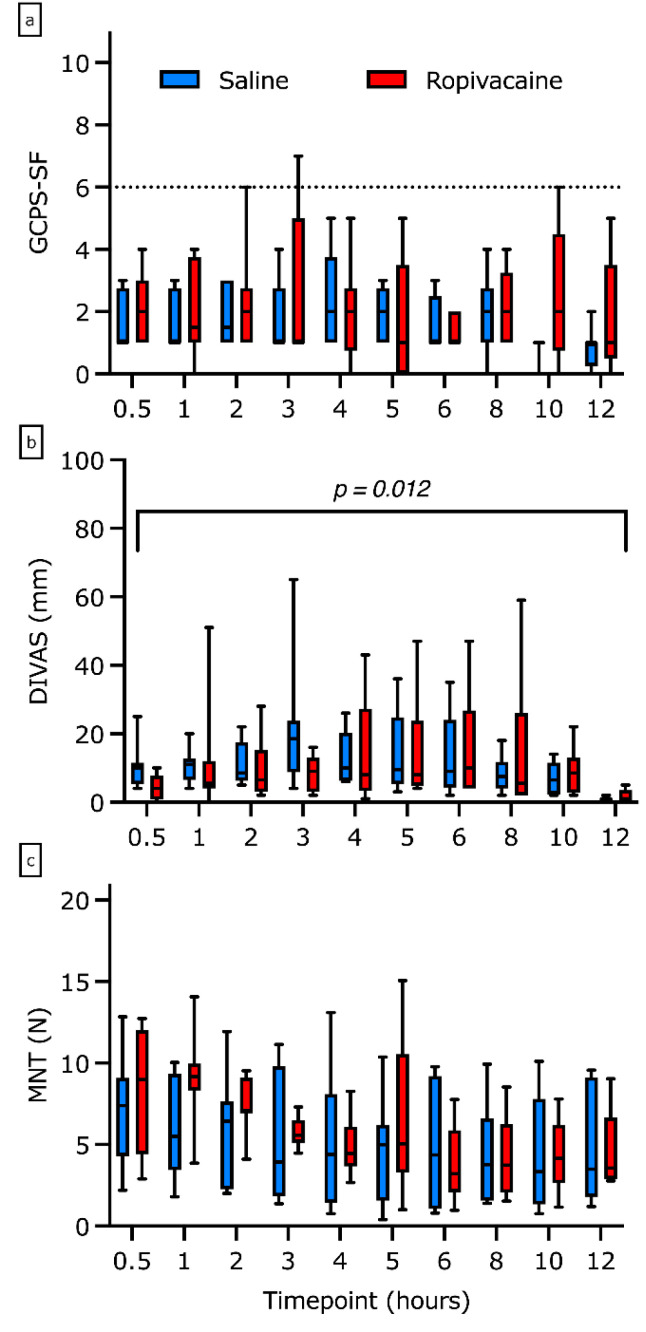
(**a**–**c**). Pain scores as attained with (**a**) the Short Form of the Glasgow Composite Pain Scale (GCPS-SF), (**b**) the dynamic interactive visual analogue scale (DIVAS), and (**c**) values for mechanical nociceptive threshold testing (MNT) of eight dogs in group Control and group Ropivacaine, respectively, represented as boxplots. The median and the upper (75%) and lower (25%) quartile are depicted with the box while the whiskers indicate the range. The dotted line in the GCPS-SF diagram represents the cutoff value at which rescue analgesia was administered.

**Table 1 animals-13-01489-t001:** Median and range of age, bodyweight, duration of anaesthesia, duration of surgery, ratio of the length of incision (LOI) to the xiphoid-to-pubis distance (XTP), and the numerical rating scale (NRS) of anticipated pain intensity as estimated by the surgeon, in 16 clinical canine patients undergoing major abdominal surgery and receiving either ropivacaine 2 mg/kg intraperitoneally and 1 mg/kg incisionally (group Ropivacaine), or an equal volume of isotonic saline (group Control). Eight dogs were included in each group. The duration of anaesthesia is the measured time between administration of induction agent and cessation of administration of anaesthetic gas. The duration of surgery is the time between skin incision and last suture. The ratio of LOI to XTP was calculated by LOI divided by XTP. The NRS was asked on a scale on which 0 indicated absence of any pain, and 10 indicated the worst pain imaginable.

	Group Ropivacaine	Group Control	*p*-Value
Age (months)	75 (9–135)	59 (6–122)	0.77
Bodyweight (kg)	13 (6.8–30)	9 (2.5–33)	0.94
Duration of anaesthesia (minutes)	134 (106–170)	124 (111–208)	0.32
Duration of surgery (minutes)	70 (50–85)	66 (43–123)	0.52
Ratio LOI: XTP (%)	56 (36–71)	55 (32–85)	0.85
NRS of anticipated pain intensity (surgeon)	3.5 (3–5)	3 (3–4)	0.23

LOI = length of incision, XTP = xiphoid-to-pubis distance, NRS = numerical rating scale.

**Table 2 animals-13-01489-t002:** Individual description of 16 clinical canine patients undergoing major abdominal surgery and receiving either ropivacaine 2 mg/kg intraperitoneally and 1 mg/kg incisionally (group Ropivacaine (R)), or an equal volume of isotonic saline (group Control (C)). Breed and intervention are indicated for all dogs as well as drugs for premedication including their route of administration, and drugs for induction which were all administered intravenously. The numbers written in bold letters indicate the dogs that received rescue methadone.

Dog N° and Group	Breed	Premedication		Induction			Procedure
		FEN (µg/kg)	DEX (µg/kg)	PRO (mg/kg)	ALF (mg/kg)	MIDA (mg/kg)	
1C	Irish Soft Coated Wheaten Terrier	4.0	5.0 IV	2.0	-	0.2	Partial pancreatectomy
2C	Chihuahua	5.0	2.0 IV	1.0	-	0.2	Nephrectomy
**3R**	Mixed breed	5.0	-	2.5	-	0.2	Adrenalectomy
**4R**	German Pinscher	3.0	5.0 IM	-	0.25	0.2	Splenectomy, Liver biopsies
5R	Dachshund	5.0	2.0 IV	1.0	-	0.2	Cystotomy
6C	Podenco Ibicenco	5.0	1.0 IV	1.0	-	0.2	Splenectomy, Liver biopsies
7C	Mixed breed	5.0	2.0 IV	0.75	-	0.2	Ureteroneostomy
8R	Miniature Schnauzer	5.0	-	3.0	-	-	Closure of portosystemic shunt, Ovariectomy
9C	Havanese	5.0	-	-	2.0	-	Closure of portosystemic shunt
**10R**	Mixed breed	5.0	4.0 IM	2.0	-	-	Closure of portosystemic shunt
11C	Mixed breed	5.0	5.0 IM	2.0	-	-	Closure of portosystemic shunt
12R	Basset	-	5.0 IM	0.75	-	0.1	Cholecystectomy
13C	Mixed breed	2.0	0.5 IV	2.0	-	0.2	Splenectomy
14R	Boxer	5.0	0.5 IV	-	1.25	0.2	Nephrectomy
15R	West Highland White Terrier	5.0	2.0 IV	-	0.5	0.2	Ovariohysterectomy (metropathy)
16C	Weimaraner	5.0	2.0 IV	-	0.75	0.2	Enterotomy

ALF = alfaxalone, DEX = dexmedetomidine, FEN = fentanyl, IM = intramuscular, IV = intravenous, MIDA = midazolam, PRO = propofol.

## Data Availability

The data presented in this study are available in detail on request from the corresponding author. The detailed data are not publicly available due to privacy restrictions.

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
