# Peer review of "Evaluation of the Analgesic Efficacy of Undiluted Intraperitoneal and Incisional Ropivacaine for Postoperative Analgesia in Dogs after Major Abdominal Surgery"

_animals, 2023, doi:10.3390/ani13091489_

Round 1
Reviewer 1 Report (Previous Reviewer 4)
I am satisfied with the extensive corrections done by the authors and my queries/clarifications have been nicely addressed and implemented by the authors and well explained. I admire the extensive efforts put on by the authors to improve the quality of the manuscript. Insertion of figure 1 to explain the protocol will definitely help the readers understanding the protocol.
Quality of english language is fine and almost flawless. The flow of the language is quite scientific in the material and methods section and results. The introduction and discussion will keep the readers interested in the study.
Author Response
Dear Reviewer 1,
thank you very much for your positive and kind response to our revisions. We are happy to have met your expectations and would like to thank you once again for your valuable time, effort and work as a reviewer which helped us to improve the quality of our manuscript.
Kind regards,
the authors
Reviewer 2 Report (Previous Reviewer 1)
The manuscript greatly improved.
However, I just have some small remarks:
Line 153: replace "the cone" with "the tip" of the syringe.
Line 332: replace "done in" with "conducted on"
Line 333: replace "and to stay on the safe side" with "and to limit the risk of overdose"
Line 333: replace "aim" with "intended"
Line 398: You may remove "in the worst case after patients undergoing elective procedures, such as ovariectomy, have been discharged"
I have no further remarks related to the manuscript
Thank you!
The structure of phrases may be improved by cutting them into shorter sentences
Author Response
Dear Reviewer 2,
thank you very much for your positive comments. We are happy to have implemented all your suggestions in the respective lines which should be easy to find. In addition, according to your advice, we cut some long sentences into shorter ones to improve the structure and to facilitate reading. These changes can be found in lines 17-20, 153-155, 202-205, and 377-381 (all marked by track changes). Once again, thank you very much for another review of our manuscript, and to help us improve it with your valuable time and work.
Kind regards,
the authors
Line 153: replace "the cone" with "the tip" of the syringe.
Thank you, this has been done (line 153): “The tip of the syringe containing the drug was inserted before the last sutures were performed.”
Line 332: replace "done in" with "conducted on"
This has also been changed to (line 334): “As in the present study investigations were conducted on clinical patients, and to limit the risk of overdose, the authors did not intend to exceed a maximum dose of 3 mg/kg, leading to an IP dose of 2 mg/kg combined with an INC dose of 1 mg/kg.”
Line 333: replace "and to stay on the safe side" with "and to limit the risk of overdose"
Thank you, see above (now starting in line 334).
Line 333: replace "aim" with "intended"
Thank you, this is also implemented in the sentence above (starting in line 334).
Line 398: You may remove "in the worst case after patients undergoing elective procedures, such as ovariectomy, have been discharged"
Thank you, this part of the sentence has been removed. The remaining part is (line 400): “Possible neurological toxicity signs due to systemic absorption could be masked by sustained sedation after anaesthesia and might only become obvious well after recovery.”
Reviewer 3 Report (Previous Reviewer 2)
An important addition to the veterinary literature and will add to patient comfort post-surgery
Author Response
Dear Reviewer 3,
thank you very much for your positive comment, and for taking the time to review our manuscript again. We are convinced that its quality improved a lot, and that is possible through the valuable work of you and the other reviewers. Thanks again!
Kind regards,
the authors
Reviewer 4 Report (Previous Reviewer 3)
As expressed previously, the paper had already been well worked out and presented. It seems to me that the extra suggestions have been taken on board and added to the text.
The reading is fluent and pleasant.
Congratulations
Author Response
Dear Reviewer 4,
thank you very much for reviewing our manuscript again, and for your positive and kind comments. We are convinced that the quality of our manuscript improved a lot thanks to your suggestions and work on it, and we would like to thank you once again for all your time and effort to do the valuable work of reviewing.
Kind regards,
the authors
This manuscript is a resubmission of an earlier submission. The following is a list of the peer review reports and author responses from that submission.
Round 1
Reviewer 1 Report
Thank you so much for submitting this article.
Please find below my remarks:
line 114 & 117 - why did the dosage of dexmedetomidine and fentanyl vary?
line 114 - For animals that received dexmedetomidine IM to facilitate the IV line placement, at what interval was the IV fentanyl and dexmedetomidine administered as premedication? Did the dexmedetomidine IM influence the analgesic protocol of the procedure?
line 128-131 - on which criteria the fentanyl dose rates were set? Furthermore, does carprofen and metamizole produce the same postoperative analgesic effects (there is at least one article published already)? Both have a profound impact on your data and your final conclusions.
line 147 - the abdominal surgeries produce both somatic and visceral pain. And so, the type of surgery (and not only the size of the midline incision) is also important.
line 162 - Why is pethidine administered to all patients (and not only as rescue analgesia) if the objective of the study was to assess the postoperative analgesic effects of a regional anaesthetic technique.
line 167 - Postoperative dexmedetomidine is influencing your results and the subject should have been excluded from the study
line 175 - How come that an abdominal surgery may impaire mobility?
line 212 - However the dose of fentanyl and dexmedetomidine varied between individuals and groups, making the results inconclusive/difficult to interpret. Furthermore, varying the type of procedure (type of surgery) produces inconclusive results too. It is difficult, if not impossible, to interpret the postoperative analgesic effect of a protocol when comparing a patient which suffered a partial pancreatectomy compared to one suffering a cystotomy.
The idea of your study is great and such studies are strongly encourage. However, I would advice you to change the study design, use the technique only for single type procedures/surgeries, and use a standardise analgesic protocol for all animals included in the study. OR, you may increase the population to make this study valid in the actual form. Until then, I am obliged to reject the publication of this article.
Reviewer 2 Report
Important topic.
Line 112 aseptic (replace aseptical)
Line 122, 122 boluses (preferred to boli)
Line 142-143 The anesthetic inhalant concentration was titrated to the appropriate depth based on clinical signs
Line 311 humans (preferred to people)
Line 317 unlikely (vs improbable)
Line 321-322 must be considered in future (further) dose researching (finding) studies
Line 422 determine a more effective (sensible) dosing
Line 448-449 The APC was funded by XXX. Why is the funding agency not listed?
Questions to the author
As previous studies of ropivacaine with higher dosages used (4.4+mg/kg) than those in the current study (3mg/kg) had no effect, why was a lower dose selected?
Line 379-383 Why was incisional Ropivicaine administered post-operatively rather than pre-op as previously known ,recommended and commonly used in practice? There are many studies describing this. The following cites many studies along with recommendations: Grubb, T, Lobprise, H. Local and regional anaesthesia in dogs and cats: Descriptions of specific local and regional techniques (Part 2). Vet Med Sci. 2020; 6: 218– 234.
Line 413 assessor of pain was a relatively inexperienced veterinary master student. Is this a post-graduate graduate in a MSc programme?
Reviewer 3 Report
The paper is well structured. It is clear and pleasant to read. I did not judge the language as I am not a native speaker, but the text was easy to understand. Below are a few considerations that I hope will be useful in improving the paper.
Line 9: “Undertreatment of pain in animals is an important problem as animals”
Line 9: The word 'animals' is repetitive. I would suggest reworking the frease
Section material and methods
The description of the anaesthesiological protocol provides for a different intra-operative fentanyl dosage range and this is clearly understandable according to the patient's needs. The different surgical procedures presumably had different intra-operative nociceptive management. Did you find any significant differences in intra-operative management that may have influenced post-operative pain management? How did you ensure that you had the same baseline condition in post-operative pain perception?
A dog benefited from carprofen and all the others from metamizole. Do you think that this case which was not 'standardised' with the other cases had any influence on the results? Did you think that you could exclude it from the data collection in order to risk it influencing the work?
The effectiveness of local anaesthetic administration in the abdomen is controversial. Indeed, in the discussion section you argue this topic very richly. Concerning the administration of the anaesthetic in the abdomen, it is unclear whether the anaesthetist performed the drug delivery according to a standardised criterion for each dog, I would consider adding this.
Line 147 and 160: Line 147 speaks of surgeons in the singular and line 160 speaks of surgeons in the plural. is this a personal misinterpretation? If so, I would suggest explaining the concept better
For post-operative pain, 6 hours after surgery, physiological pain reduction over time and the return to the pre-operative pain scores within a few hours, a phenomenon already reported after ovariohysterectomy in bitches, regardless of the analgesic protocol used McKune, C.M.; Pascoe, P.J.; Lascelles, B.D.X.; Kass, P.H. The challenge of evaluating pain and a pre-incisional local anesthetic block. PeerJ 2014, 2, e341
Section discussion is very rich in interesting topics. I would suggest dividing it into sections to make it more reader-friendly.
Reviewer 4 Report
Reviewer comments for manuscript ID animals-2241799 entitled ‘Evaluation of the analgesic efficacy of undiluted intraperitoneal and incisional ropivacaine for postoperative analgesia in dogs after major abdominal surgery’
General Comments
The study is a promising effort to establish local anaesthetics a vital ingredient of a multimodal post operative analgesic protocol. It is an endeavour of clinicians and anaesthesiologists to minimise post operative surgical pain for faster and complication free recovery in veterinary patients. Local anaesthetics have been routinely used as pre, intra and post-operative pain management in canines and equines especially through intra peritoneal and incisional infiltration routes.
The manuscript is nicely presented with minimal errors in writing. The data has been objectively analysed using appropriate statistical methods. The gaps have been identified in literature and the discussion is comprehensive analysing the results and identifying the limitations of the study. I have doubts over the selection of protocols for the treatment /anaesthetic groups versus the control groups. There is no uniformity in these groups that makes the comparison of treatment effects difficult. This is a serious limitation of the study even more than the lower sample size. The various analgesic protocols might confuse the reader and this approach is scientifically flawed. I do understand the clinical nature of the study but uniformity in the treatments groups is mandatory for comparison with control groups. I recommend exhaustive revision of the study before I recommend the publication of the manuscript.
Specific Comments
Line 11: Please replace ‘be it for spay of healthy bitches or for curative reasons’ with ‘be it elective or for curative reasons’.
Line 72 : should be ‘recommends’ instead of ‘claims’. Please clarify.
Line 100: Please replace ‘could be’ with ‘were’
Line 104: Please replace ‘dogs’ with ‘ones’
Lines 109-32: Please draw a chart of the anaesthetic protocol followed for easy comprehension of the reader. It looks confusing here and I am not able to differentiate the anaesthetic groups.
Line 340-43: Please insert a reference in support of this assertion.